# LEF1 enhances *β*-catenin transactivation through IDR-dependent liquid–liquid phase separation

Bing Zhao[1], Zhuoxin Li[1], Shaoqing Yu[2], Tingting Li[1,3], Wen Wang[1], Ran Liu[1], Biyu Zhang[1], Xiya Fang[1], Yezhuang Shen[1], Qiuying Han[1,3], Xin Xu[1,3], Kai Wang[1,3], Weili Gong[1], Tao Li[1,3], Ailing Li[1], Tao Zhou[1,3], Weihua Li[1,]*, Teng Li[1,3,]*

Wnt/*β*-catenin signaling plays a crucial role in cancer development, primarily activated by *β*-catenin forming a transcription complex with LEF/TCF in the nucleus and initiating the transcription of Wnt target genes. Here, we report that LEF1, a member of the LEF/TCF family, can form intrinsically disordered region (IDR)-dependent condensates with *β*-catenin both in vivo and in vitro, which is required for *β*-catenin-dependent transcription. Notably, LEF1 with disrupted IDR lost its promoting activity on tumor proliferation and metastasis, which can be restored by substituting with FUS IDR. Our findings provide new insight into the essential role of liquid–liquid phase separation in Wnt/*β*-catenin signaling and present a potential new target for cancer therapy.

## Introduction

Canonical Wnt/*β*-catenin signaling is crucial for determining cell fate, promoting proliferation, facilitating development, and aiding in tissue repair (Clevers, 2006; Clevers & Nusse, 2012; Bugter et al, 2021). Aberrant Wnt/*β*-catenin signaling can lead to developmental abnormalities and diseases such as cancer and diabetes. However, *β*-catenin, which dominates the expression of genes in Wnt signaling, has no intrinsic ability to bind DNA (Morris et al, 2010; Nusse & Clevers, 2017; Bugter et al, 2021). Upon Wnt signaling activation, *β*-catenin translocates into the nucleus where it binds to the N-terminus region of TCF/LEF transcription factors and recruits co-factors that facilitate the transcription of target genes (Behrens et al, 1996; Tutter et al, 2001).

Lymphoid enhancer factor 1 (LEF1) is a member of the TCF/LEF subfamily that functions as a transcription factor and key mediator of Wnt signaling throughout metazoans (Behrens et al, 1996; Aulicino et al, 2020). Dysregulation of LEF1 has been observed in various cancer cells, suggesting its potential as a marker for cancer diagnosis and prognosis (Santiago et al, 2017). Previous studies have primarily focused on elucidating the stabilization and nuclear accumulation of *β*-catenin, but the mechanism underlying the co-activation of

transcription by *β*-catenin and LEF1 remains incompletely understood (Griffin et al, 2018; Imamoto, 2018; Anthony et al, 2020).

Recent research has demonstrated that liquid–liquid phase separation (LLPS) orchestrates numerous biomedical reactions in cells (Alberti & Carra, 2018; Boeynaems et al, 2018; Boija et al, 2018). Phase separation induces the formation of membrane-less organelles at the micron scale, which function to concentrate multiple proteins and RNA (Murray et al, 2017; Wang et al, 2018). LLPS has been reported to play multiple roles in regulating gene transcription, RNA metabolism, cell division, and synapse function (Ong & Torres, 2020; Wang et al, 2021a). Previous studies have established that proteins undergoing LLPS often possess intrinsically disordered regions (IDRs), which are considered distinctive and common characteristics of eukaryotic proteins (Haynes et al, 2006). Proteins containing IDR are enriched in numerous biomolecular condensates, such as P bodies, stress granules, PML bodies, and various nuclear structures.

Here, we demonstrate that LEF1 undergoes condensation into droplets with *β*-catenin upon Wnt signaling activation in colorectal cancer cells. We discovered that the IDR of LEF1 is essential for its LLPS both in vivo and in vitro. Furthermore, phase separation mediated by the IDR domain is necessary for *β*-catenin and LEF1-targeted gene transactivation. Meanwhile, we found that the interaction between LEF1 and *β*-catenin plays an important role in condensate formation for both LEF1 and *β*-catenin. Importantly, LEF1 promotes the proliferation and migration of colorectal cancer cells in an IDR-dependent manner. Therefore, our findings provide insights into a potential mechanism for Wnt-mediated tumor proliferation via LEF1 LLPS.

## Results

### LEF1 undergoes phase separation in vivo upon Wnt activation

In our previous studies, we noticed that LEF1 exhibited punctate distribution in SW480, Jurkat, and HEK293T cells when expressed at high levels. This is reminiscent of proteins forming condensates in cells via LLPS (Ahn et al, 2021). Therefore, we observed the cellular behavior of LEF1 through immunofluorescence in SW480 cells to

[1]National Center of Biomedical Analysis, Beijing, China   [2]School of Clinical Medicine, Tsinghua University, Beijing, China   [3]Nanhu Laboratory, Jiaxing, China

Correspondence: tnli@ncba.ac.cn; whli@ncba.ac.cn
*Weihua Li and Teng Li jointly supervised this work

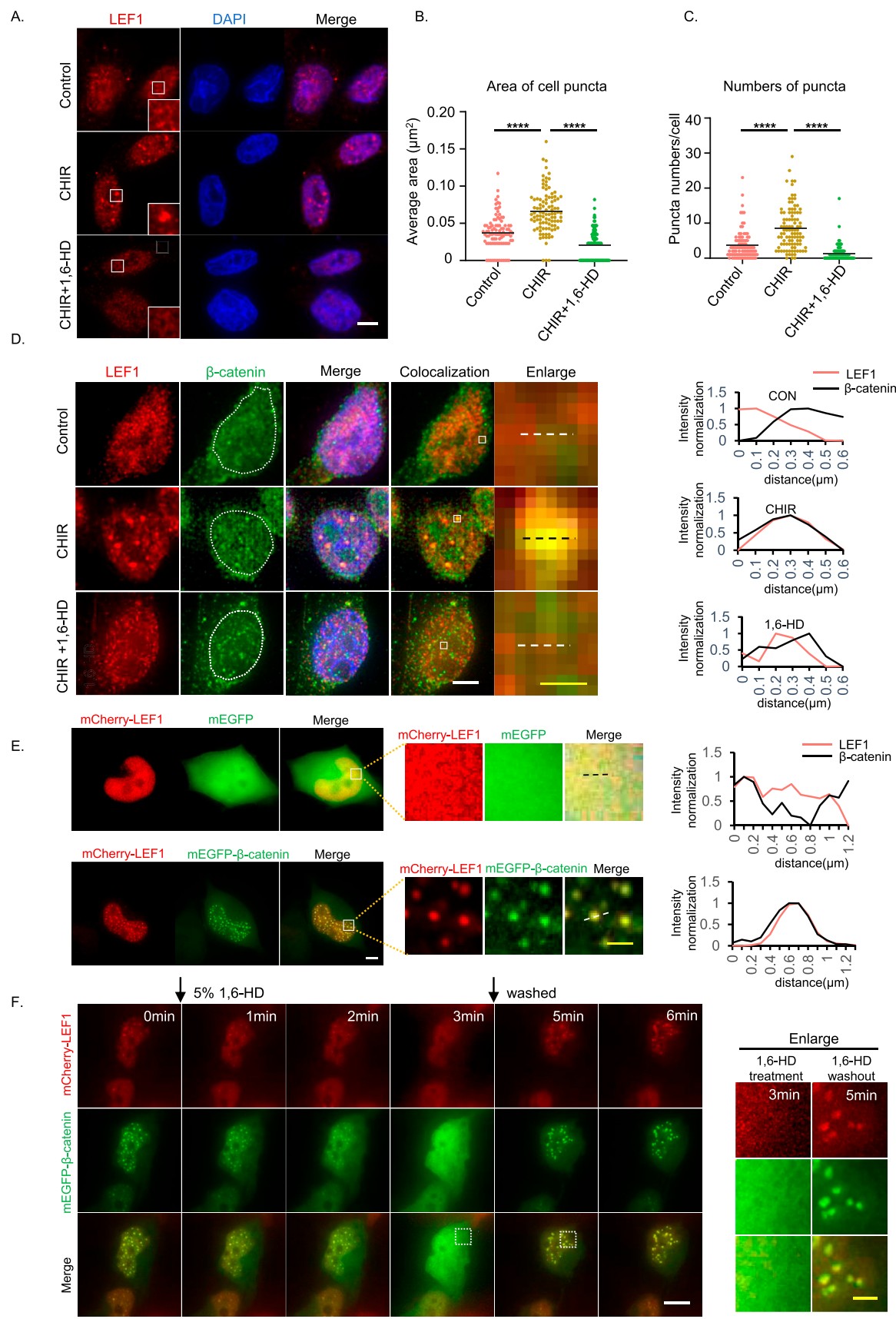

investigate whether LEF1 facilitates β-catenin–mediated trans-activation through LLPS. We observed an increase in the size of LEF1 puncta in the nucleus of SW480 cells after treatment with CHIR99021, a Wnt pathway activator. In addition, these puncta disappeared after treatment with 1,6-hexanediol (1,6-HD), a chemical commonly used to disrupt condensates (Fig 1A–C) (Pak et al, 2016; Sabari et al, 2018). Another important finding was that β-catenin colocalized in LEF1 puncta, whereas accumulating in the nucleus upon Wnt pathway activation. The colocalized puncta also disappeared after treatment with 1,6-HD (Fig 1D). These data suggest that LEF1 could undergo LLPS with β-catenin upon Wnt activation.

To validate the condensates of LEF1 under Wnt signaling stimuli, we overexpressed β-catenin in HEK293T cells to activate Wnt signaling with LEF1 overexpression. We observed LEF1 puncta when co-expressed with β-catenin, and the puncta contained both LEF1 and β-catenin (Figs 1E and S1A). In addition, we observed that endogenous LEF1 was also co-localized into puncta formed by overexpressed β-catenin (Fig S1B). After treatment with 1,6-HD, these puncta rapidly disappeared and quickly recovered within minutes upon removal of the 1,6-HD (Fig 1F). Besides, we used FRAP to monitor the molecular dynamics of LEF1 and β-catenin within the droplets (Fig S1C and Video 1). Both LEF1 and β-catenin molecules in the bleached area were actively exchanged with surroundings, as indicated by recovery after photobleaching. In conclusion, our results suggest that LEF1 undergoes LLPS with β-catenin in vivo in response to Wnt-signaling activation.

## LEF1 undergoes phase separation with β-catenin in vitro

To investigate whether LEF1 can form phase-separated droplets with β-catenin, we purified recombinant mCherry–LEF1 and mEGFP–β-catenin to perform a phase-separation assay in vitro (Fig S2A and B). The proteins were diluted to varying concentrations in buffers with 10% PEG-6K added as a crowding agent to simulate cellular conditions. Both mCherry–LEF1 and mEGFP–β-catenin formed droplets in a concentration-dependent manner (Figs 2A, B, and F and S2C), whereas the behavior of mEGFP–β-catenin was consistent with previous reports (Zamudio et al, 2019). Further characterization of these droplets revealed their sensitivity to salt concentration and susceptibility to 1,6-HD (Fig 2C and D). Besides, time-lapse imaging results indicated that these droplets readily fused into larger structures over time (Fig 2E and Video 2). These dynamic reversible processes observed are consistent with protein behavior during LLPS.

We then investigated whether LEF1 and β-catenin can form co-condensates and if their phase behavior is regulated by protein concentration. When we mixed β-catenin and LEF1 in solution, new droplets

consisting of both proteins formed and enlarged as the protein concentration increased (Fig 2F). In summary, our findings demonstrate that LEF1 undergoes phase separation with β-catenin in vitro.

## LEF1 LLPS and Wnt target gene activation are dependent on IDR

It has been reported that the HMG domain of LEF1 induces DNA bending, serving as an architectural element that potentially facilitates communications between distantly located protein-binding sites through precise spatial alignment (Giese et al, 1992). However, the mechanism by which LEF1 and other proteins assemble into a multiprotein complex to regulate target genes remains unknown. To address this question, we next sought to find out whether LEF1 exerts its transcriptional activation function through LLPS. Considering the ability of IDR to facilitate LLPS through various weak interactions (Shimobayashi et al, 2021; Hirose et al, 2022), we used PONDR algorithms (http://pondr.com/) to predict the presence of IDR in LEF1's protein sequence. The analysis revealed a predominantly unfolded IDR within LEF1's trans-activation domain (Fig 3A). To further investigate the contributions of IDR in LEF1 phase-separation behavior, we divided the IDR region of LEF1 into three parts: IDR1 from amino acids 81 to 160 aa, IDR2 from 161 to 240 aa, and IDR3 from 241 to 310 aa. We created a series of constructs by truncating the IDR while retaining their β-catenin–binding region and DNA-binding region (Fig 3B). The immunoprecipitation assay confirmed that all the mutants of LEF1 retained their β-catenin–binding ability (Fig 3F).

A dual-luciferase reporter gene assay was performed in HEK293T cells to detect the function of LEF1 and its truncated forms in β-catenin dependent transactivation. The results showed that full-length LEF1 enhanced luciferase activity, but none of the IDR-deleted truncated forms (Fig 3C), suggesting that IDR is necessary for the transcriptional activity of LEF1. Consistently, in HEK293T cells overexpressing β-catenin with LEF1 or truncated forms, the formation of puncta was significantly suppressed under IDR depletion compared with full-length LEF1, even when only 1/3 of the IDR was missing (Fig 3D and E). The β-catenin–binding domain (1–80 aa) of LEF1 is known to be essential for canonical Wnt activation; however, its role in the formation of condensates between LEF1 and β-catenin remains unknown. We investigated the localization of a LEF1 mutant with a deletion in its β-catenin–binding region and observed that the deletion mutant exhibited a significantly reduced ability to form condensates. Furthermore, overexpression of the LEF1 mutant without the β-catenin–binding domain led to a decrease of β-catenin puncta in the nucleus (Fig S3A and B). These findings suggest that the interaction between LEF1 and β-catenin plays an important role in condensate formation for both proteins.

**Figure 1. LEF1 undergoes phase separation in vivo upon Wnt activation.**
**(A)** Immunofluorescence for LEF1 in SW480 cells when Wnt signaling is activated by CHIR99021 for 24 h. Cells were treated with 5% 1,6-HD for 10 s. Scale bar, 5 $\mu$m. **(B)** Dots show the average area of puncta from individual cells and mean values (black horizontal lines). 100 cells were analyzed in each group (****$P < 0.0001$, one-way ANOVA). **(C)** Dots show puncta numbers of SW480 cells after different treatments and mean values (black horizontal lines). 100 cells were analyzed in each group (****$P < 0.0001$, one-way ANOVA). **(D)** Left, immunofluorescence for LEF1 and β-catenin when Wnt signaling activated with or without 5% 1,6-HD for 10 s. Scale bar, 5 $\mu$m (white horizontal line), 0.5 $\mu$m (yellow horizontal line). Right, normalized fluorescence intensity profiles of dotted lines are indicated in the images on the left. **(E)** Left, cells were transfected with mEGFP–β-catenin and mCherry–LEF1 for 24 h and imaged. Scale bar, 5 $\mu$m (white horizontal line), 1 $\mu$m (yellow horizontal line). Right, normalized fluorescence intensity profiles of dotted lines are indicated in the images on the left. **(F)** Time-lapse imaging of cells transfected with mEGFP–β-catenin and mCherry-LEF1 after treatment with 1,6-HD and washout. Scale bar, 5 $\mu$m (white horizontal line), 2 $\mu$m (yellow horizontal line).

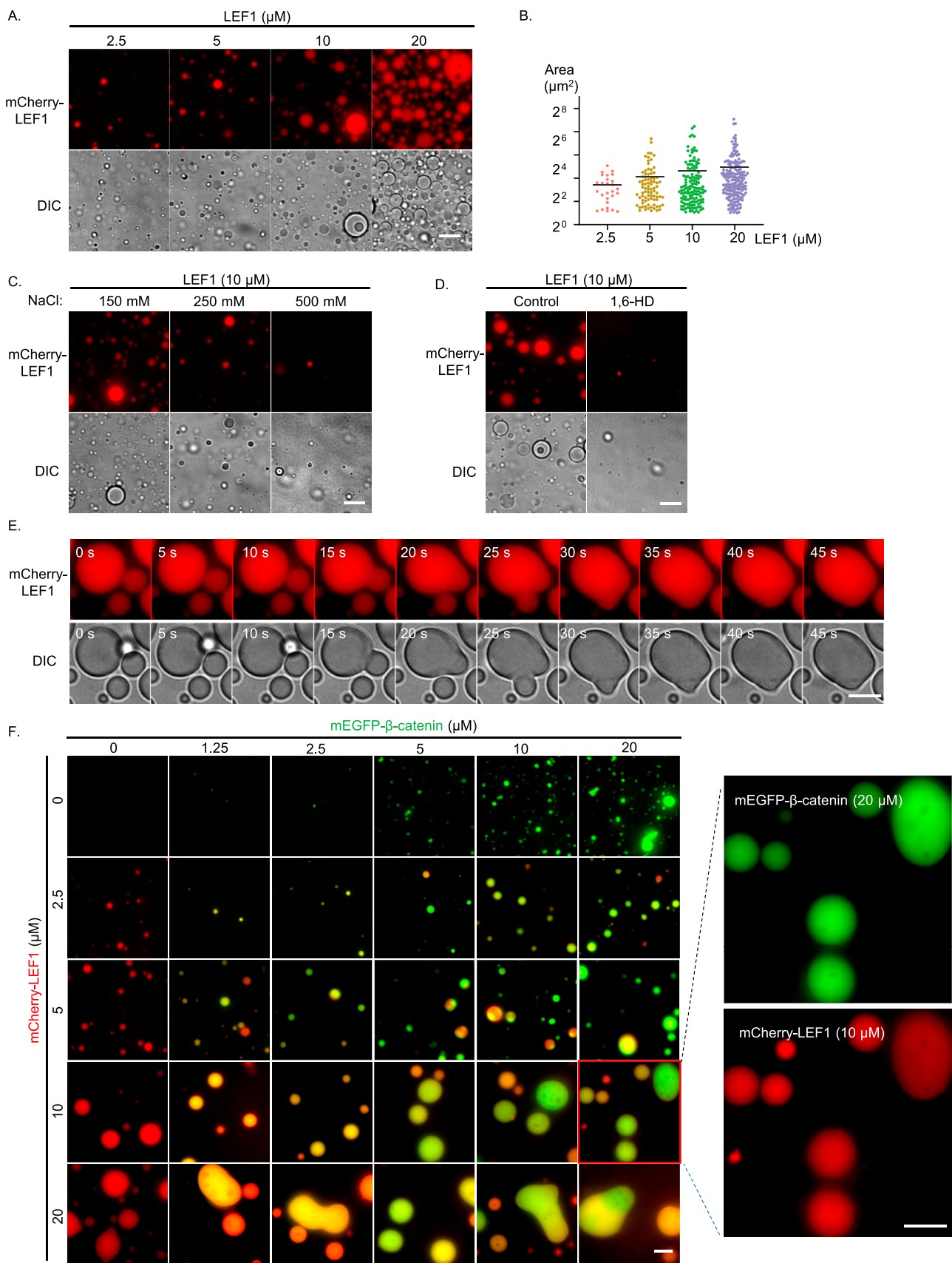

We further explored the phase separation of LEF1 IDR-depleted mutants in vitro. Truncation of IDR made it difficult to observe droplets compared with full-length LEFI (Fig 3G), which is consistent with the phenomenon in vivo. When mixed with β-catenin at the same concentration, LEF1 formed larger droplets than IDR-deficient mutants (Fig 3H). These results above suggest that IDR-dependent phase separation of LEF1 is correlated with its transcription activity and puncta size in cells.

### IDR-dependent phase separation of LEF1 is indispensable for its transcriptional activation

As the LEF1 IDR coincides with its transactivation domain, we replaced the LEF1 amino acid with the FUS-IDR domain (Qamar et al, 2018) (LEF1 81–160 aa to FUS 27–107 aa and LEF1 81–240 aa to FUS 1–162 aa) to clarify whether its transactivation is mainly driven mainly by LLPS (Figs 4A and S4A). The Luciferase reporter assay revealed that the IDR substitution of LEF1 reinstated the transcriptional activation function that was abolished by depleting LEF1 IDR (Fig 4B). When overexpressed with β-catenin in HEK293T cells, the IDR substitute mutants of LEF1 formed puncta in the nucleus similar to full-length LEF1 (Fig 4C and D). IDR substitution also restored droplet formation comparable with full-length LEF1, both individually and together with β-catenin (Figs 4E and F and S4B) in vitro. These findings further underscore the essential role of IDR-driven phase separation in the transcriptional activation of LEF1.

### LEF1 phase separation promotes tumor growth and metastasis

Next, we analyzed LEF1 expression levels in tumor and normal samples obtained from The Cancer Genome Atlas and Genotype-Tissue Expression data through the GEPIA platform (http://gepia.cancer-pku.cn/). Our findings revealed a significantly elevated expression of LEF1 in various tumors compared with normal tissues, including melanoma, thymic cancer, and colorectal cancer (Fig 5A). Subsequently, we analyzed the correlation between β-catenin and LEF1 in colorectal cancer. The results of the correlation analysis indicated a positive association between the expression levels of β-catenin and LEF1 in colorectal cancer (Fig S4C).

Therefore, we stably transfected LEF1 and its mutants, respectively, in HCT116 cells, a colon cancer cell line expressing low levels of LEF1, to investigate the role of LEF1 LLPS in tumor proliferation and migration. We found that full-length expression of LEF1 promotes clone formation of HCT116 cells, but not mutants with deleted IDR. The IDR substitute mutant of LEF1 exerted the same promotion effect as full-length LEF1 (Fig 5B and C). To confirm whether the IDR of LEF1 contributes to clone formation through β-catenin–mediated transcription of target genes, we assessed the

transcription activity of *c-jun* and *c-myc* under the expression of LEF1 WT or IDR-deletion mutants. Our results indicate that only LEF1 WT, but not phase separation-deficient mutant, can promote tumor cell growth by the transcription of downstream target genes (Fig S4D). Furthermore, we evaluated cell migration using a transwell assay. As shown in Fig 5D, the migration of HCT116 cells was promoted after transfection with LEF1 or IDR substitute mutant transfection, whereas IDR deletion mutant-expressing cells did not show significant changes. These findings suggest that LEF1 promotes the proliferation and migration of tumor cells through its IDR-mediated phase separation.

## Discussion

Our results demonstrate that LEF1 can undergo LLPS with β-catenin when Wnt/β-catenin signaling is activated in tumor cells. We further confirmed the phase separation of LEF1, either alone or with β-catenin in vitro. Importantly, we discovered that the IDR of LEF1 is necessary for Wnt/β-catenin signaling transactivation and LEF1/β-catenin co-condensate formation both in vivo and in vitro. In addition, the interaction between LEF1 and β-catenin is essential for their co-condensate formation in vivo. Furthermore, disruption of IDR in LEF1 abolishes its capacity to promote tumor proliferation and metastasis, whereas substitution with FUS IDR reinstates this promoting activity.

LEF1 plays a central role in the Wnt/β-catenin signaling pathway as a mediator and regulates downstream cellular effects, making it crucial for controlling protein expression. Studies have shown that complexes formed by β-catenin and TCF/LEF proteins, along with additional regulatory factors, induce specific transcriptional responses in different cell types (Boija et al, 2018; Chong et al, 2018). Our findings propose a model in which LEF1 acts as a scaffold for the assembly of other proteins into a multiprotein complex via LLPS (Fig 5E). Furthermore, our results confirm that transcription factors facilitate the compartmentalization of transcription regions and regulate transcription through LLPS.

As a well-documented cancer driver, Wnt/β-catenin signaling has been extensively studied. Despite numerous small molecules blocking Wnt signaling in various cell models (Wang et al, 2021b), no Wnt inhibitor has advanced to clinical use. Furthermore, although many Wnt inhibitors have been identified via high-throughput screening, β-catenin was rarely shown to be a direct target (high-throughput screening) (Cui et al, 2018). Therefore, targeting inhibitors directly in the downstream part of the Wnt signaling would be more likely to benefit cancer patients (Emami et al, 2004; Gonsalves et al, 2011; Kahn, 2014). Targeting LEF1 IDR to disrupt phase separation may be a feasible strategy for the treatment of cancer.

---

**Figure 2.  LEF1 undergoes phase separation with β-catenin in vitro.**
**(A)** Representative images of mCherry–LEF1 droplet formation with indicated concentrations (RT, 150 mM NaCl, pH 7.5, 10% PEG-6000). Scale bar, 10 μm. **(B)** Measurement of the area and number of droplets formed in Fig 2A. **(C)** Droplet formation of mCherry–LEF1 was analyzed with different concentrations of NaCl. Scale bar, 10 μm. **(D)** Droplet formation of mCherry–LEF1 was analyzed with or without 5% 1,6-HD. Scale bar, 10 μm. **(E)** Selected frames from time-lapse movies of representative LEF1 droplets fusion. Scale bar 2 μm. **(F)** Representative images of mEGFP–β-catenin and mCherry–LEF1 mixture with the indicated module concentration. Scale bar 10 μm.

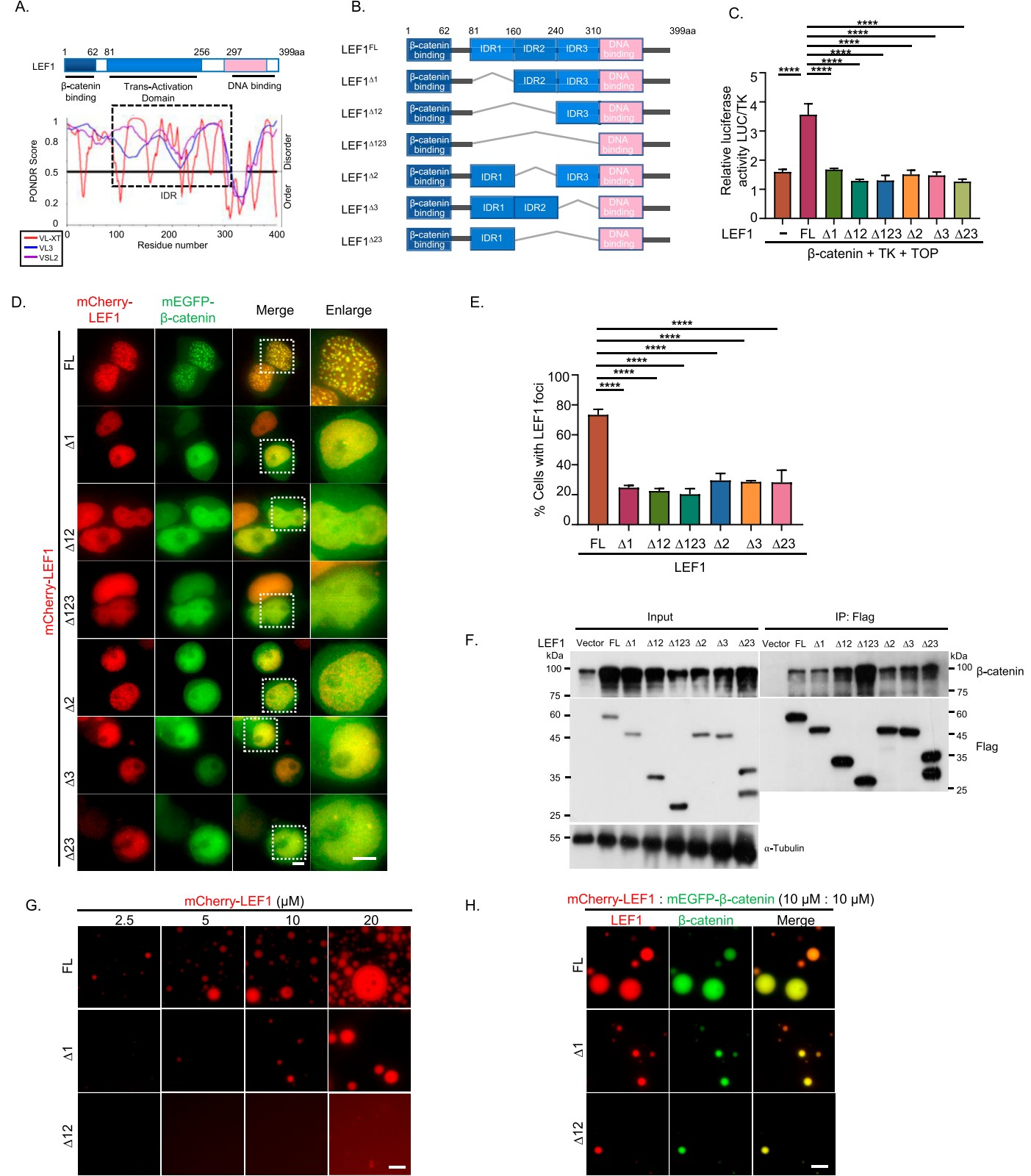

**Figure 3. LEF1 LLPS and Wnt target gene activation are dependent on IDR.**
**(A)** Domain structure and the intrinsically disordered tendency of LEF1 predicted by PONDR. PONDR assigned scores of disordered tendencies between 0 and 1 to the sequences (scores >0.5 indicates disordered). **(B)** Schematic of LEF1(FL) and truncated LEF1 with IDR depleted (Δ1, Δ12, Δ123, Δ2, Δ3, Δ23). **(C)** Activation of β-catenin with different LEF1 mutants that were detected by luciferase reporter gene assay. Error bars show the mean ± SEM from three individual experiments (****$P < 0.0001$, one-way

Targeting LEF1 IDR to disrupt phase separation may be a feasible strategy for the treatment of cancer. Meanwhile, LEF1 has been identified as a useful diagnostic marker for the development, metastases, and poor prognoses in several cancers, including chronic lymphocytic leukemia (Menter et al, 2017), breast cancer (Gilles et al, 2003), liver cancer (Singhi et al, 2014), prostate cancer (Yang et al, 2006). Therefore, detecting the level of phase separation in LEF1 may be an effective method for predicting cancer proliferation and metastasis.

LEF1 also plays a critical role in stem cell maintenance through the Wnt/$\beta$-catenin signaling, including hematopoietic (HSC), leukemic stem cells, and intestinal stem cells (Korinek et al, 1998; Willert et al, 2003; Staal & Clevers, 2005). In addition, LEF1 is implicated in the maturation of natural killer T cells and T follicular helper cells (Berga-Bolanos et al, 2015; Ortiz et al, 2015). Whether LEF1-mediated transcriptional activation regulates the above processes through LLPS needs to be further explored.

# Materials and Methods

## Cell culture, transfection, infection

HEK293T, SW480, and HCT116 cells were obtained from the Cell Resource Center, Peking Union Medical College. HEK293T cells were cultured at 37°C with 5% $CO_2$ in DMEM (Invitrogen) supplemented with 10% FBS and 1% penicillin–streptomycin, SW480 cells were cultured in IMEM (Invitrogen), and HCT116 cells were cultured in 1640 (Invitrogen) with the same additives. Sf9 cells were cultured in Sf-900 II SFM (10902096; GIBCO) supplemented with 1% penicillin–streptomycin at 28°C.

Transfection of HEK293T cells was performed using VigoFect (Vigorous Biotechnology) according to the manufacturer's instructions. The following plasmids were transfected into HEK293T cells: $\beta$-catenin-WT, LEF1-FL, and LEF1 mutants were individually constructed in the plasmid pmEGFP-C1 or pmCherry-C1. The stable HCT116 cells expressing Flag-LEF1 or Flag-LEF1-$\Delta$IDR were generated by retroviral infection followed by puromycin screening. All cell lines have been fully authenticated and tested free of mycoplasma. All plasmids used were confirmed by sequencing before use.

## Immunofluorescence

Before immunofluorescence imaging, SW480 cells were treated with 6 $\mu$m CHIR99021(Cat#04-0004; Stemgent) for 24 h in IMEM medium containing 10% FBS. For immunostaining assay, cells were extracted in CSK buffer (100 mM NaCl, 300 mM sucrose, 1 mM $MgCl_2$, 1 mM EGTA, 1 mM DTT, 10 mM PIPES pH 6.8) plus 0.1%

Triton X-100 for 5 min as described previously (Jamieson et al, 2011). Then, the cells were fixed in 4% paraformaldehyde for 10 min at RT and permeabilized with 0.5% Triton X-100 in PBS on ice for 10 min. After three washes in PBS, cells were blocked in 3% bovine serum albumin for 15 min at RT and incubated with LEF1 (CST 2230s; C12A5) and $\beta$-catenin (ab22656) antibodies in a blocking buffer at 4°C overnight. After three washes in PBS, cells were incubated in fluorescent secondary antibodies in a blocking buffer for 1 h. Next, cells were incubated with Hoechst dye for 5 min at RT after three washes in PBS. After three PBS washes, the slides were mounted with the mounting agent (Cat# ZLI-9556) and coverslips were sealed and stored at 4°C. Images were acquired using DeltaVision Deconvolution Microscope (GE Healthcare) with 60× oil immersion objective lens. Image deconvolution and analysis were operated by SoftWoRx software. Image analysis was performed using ImageJ software (National Institutes of Health).

## Immunoprecipitation and immunoblot analysis

Cells were lysed with 1 ml lysis buffer (20 mM Tris–HCl, 25 mM NaF, 2 mM EDTA, and 1% Triton X-100, pH 7.4) containing proteinase inhibitor cocktail (Cat#B14001; Bimake) for 30 min at 4°C. Protein concentrations were measured, and equal amounts of lysates were used for immunoprecipitation after centrifugation at 12 × 1,000$g$ for 10 min. Immunoprecipitation was performed with anti-FLAG M2 beads (Cat#B23102; Selleck) for 1 h at 4°C. Thereafter, the precipitants were washed five times with lysis buffer and the immunocomplexes were eluted with a sample loading buffer containing 1% SDS for 10 min at 95°C. The immunoprecipitated proteins were separated thereafter by SDS–PAGE. Immunoblotting analysis was performed with antibodies and secondary anti-mouse or anti-rabbit antibodies conjugated to HRP. Visualization was achieved with chemiluminescence.

## Protein purification

The cDNA encoding *LEF1* and its mutants, and *CTNNB1*, were cloned into a modified version of the pFastBac HT B expression vectors (10584-027; Invitrogen). All constructed vectors are sequenced for validation and transformed into DH10Bac *E. coli* (Cat#BC112-01; Biomed) for screening. Uniformly growing white colonies were selected and each monoclonal colony was added to 1 ml containing three antibiotics (50 $\mu$g/ml kanamycin, 7 $\mu$g/ml gentamicin, and 10 $\mu$g/ml). LB liquid medium was cultured at 37°C at 220 rpm. Half an hour later, PCR was used to determine whether the white spot was a positive clone.

Next, bacmid was transfected into Sf9 cells to produce a recombinant baculovirus that could express the target gene.

---

ANOVA). **(D)** Representative confocal microscopy images of HEK293T cells transfected with mEGFP–$\beta$-catenin and mCherry–LEF1 or LEF1 mutants. Scale bar, 5 $\mu$m. **(E)** A quantification of the percentage of cells that showed nuclear LEF1 puncta. Error bars show mean ± SEM from three individual experiments. (****$P$ < 0.0001, one-way ANOVA). **(F)** Immunoprecipitation and Western blot analysis of FLAG-tagged LEF1 variants co-expressed with $\beta$-catenin in HEK293T cells. **(G)** mCherry–LEF1(FL), $\Delta$1, $\Delta$12 proteins were analyzed using droplet-formation assays with the indicated module concentration. Scale bar, 10 $\mu$m. **(H)** Representative images of mEGFP–$\beta$-catenin (10 $\mu$m) and mCherry–LEF1 (FL), $\Delta$1 or $\Delta$12 (10 $\mu$m) proteins mixture in phase-separation buffer. Scale bar, 10 $\mu$m.

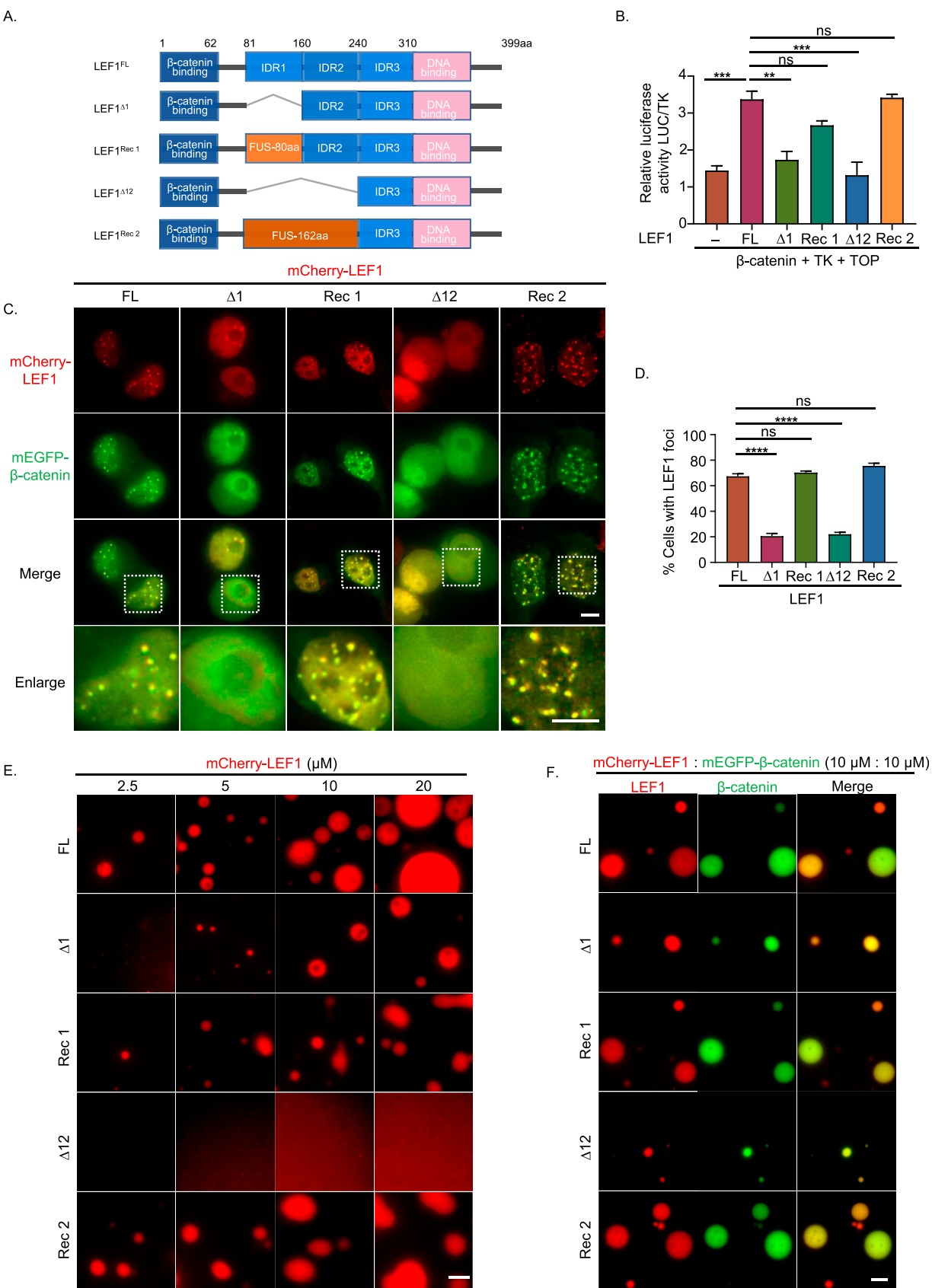

Bacmid transfections were performed using Cellfectin II Reagent (Cat#10362100; Thermo Fisher Scientific) per the manufacturer's recommendations. The transfected Sf9 cells were incubated at 28°C in an incubator. 72 h posttransfection, the cells were harvested and centrifuged at 500$g$ for 5 min. The resulting supernatant was collected and stored in the dark at 4°C as P1 virus. After confirmation of the virus titer with P1, fresh Sf9 cells were transfected with P1 and cultured to a volume of 100 ml. P2 was then collected using the previous method after 72 h, which can be used for protein purification by harvesting cells after reinfection with Sf9.

Pellets containing 500 ml of P2-transfected Sf9 cells were resuspended in 20 ml of pre-chilled lysis buffer (Tris–HCl 50 mM pH 7.5, NaCl 300 mM, glycerol 5% [vol/vol], imidazole 10 mM, DTT 1 mM, PMSF 1 mM), and placed in an ice bath for 30 min, followed by thorough mixing using a pipette. Subsequently, the initial cell lysate was sonicated to achieve complete lysis. The resulting crude lysate was then clarified through centrifugation at 12,000$g$ for 30 min before being combined with 1 ml of pre-equilibrated Ni-NTA agarose (G106-100; Genstar). The mixture was rotated for 4–5 h at 4°C. The incubated column was subsequently introduced into the rinsed protein purification gravity column, allowing the cleavage product to flow gradually under gravitational force. At this stage, the target protein became bound to the Ni-NTA agarose. As the upper liquid in the protein purification gravity column approached depletion, a wash buffer (Tris–HCl 50 mM pH 7.5, NaCl 300 mM, glycerol 5% [vol/vol], imidazole 30 mM, DTT 1 mM, PMSF 1 mM) was used to elute loosely bound proteins in a volume of 20 ml. When almost all of the wash buffer had passed through the protein purification gravity column, an elution buffer (Tris–HCl 50 mM pH 7.5, NaCl 300 mM, glycerol 5% [vol/vol], imidazole 500 mM, DTT 1 mM, PMSF 1 mM) was added to displace the target protein in a volume of 10 ml. The collected outflow was successively transferred into individual 1.5-ml tubes and analyzed using Coomassie staining on a polyacrylamide gel with a concentration of 12%.

The elution samples were concentrated to a volume of 500 $\mu$l using a 30 K Amicon Ultra-15 (UFC903096; Millipore) at 4°C at 4,000$g$/min for 30 min. Subsequently, the supernatant was transferred to an ultracentrifuge tube and centrifuged at 25,000$g$/min for 5 min at 4°C. After centrifugation, the supernatant was further purified using the GE AKTA fast protein liquid chromatography system. The collected fractions were concentrated to a final volume of ~50 $\mu$l before packaging. To prevent repeated freeze–thaw cycles, aliquots of each tube (10 $\mu$l) from each tube were snap-frozen in liquid nitrogen and stored long-term at –80°C (Alberti et al, 2018). Protein concentration was measured by NanoDrop after diluting for subsequent experiments.

## In vitro phase separation assay

Recombinant proteins were diluted to indicated concentrations in a buffer containing 50 mM Tris–HCl, 150 mM NaCl, pH 7.5, and 1 mM DTT (Cat#SH404-01; Biomed) and 10% PEG-6000 as a crowding agent. Protein solution (5 $\mu$l) was loaded onto a glass-bottom cell culture dish (Cat#801002) and imaged using a Delta Vision Deconvolution Microscope (GE Healthcare). The images presented are of droplets settled on the glass coverslip.

## Luciferase reporter assay

Typically, $2 \times 10^5$ cells per well were seeded in a 12-well plate and transfected with indicated combinations of plasmids using VigoFect (Vigorous Biotechnology). The plasmid transfection ratio was set at 2:1:1:0.1 for *LEF1*, *CTNNB1*, M50 Super 8x *TOP/FOP* Flash (LMAI Bio, LM8105, LM8196), and TK, respectively. After 24 h, cells were harvested and luciferase activities were measured using dual-luciferase reporter assay system (Cat#E1910; Promega) according to the manufacturer's instructions. All luciferase values were normalized with the activity of Renilla luciferase and represented as the mean of a triplicate set of experiments.

## Clone formation and transwell assay

Cells were digested into single cells, and cell suspension was diluted in gradient multiples. 500 cells per well were inoculated into six-well plates (Corning) and incubated at 37°C with 5% $CO_2$ for 14 d. The supernatant was discarded, and the wells were washed twice with PBS. After fixation with 4% PFA for 30 min, crystal violet dye solution was added to each well for staining for another 30 min before being slowly washed away with running water followed by air drying. Clones were counted after scanning. The cell concentration was adjusted to a final concentration of $2 \times 10^5$/ml using serum-free medium before seeding onto Transwell chambers (Cat#CLS3422-48EA; Corning). A total of 100–150 $\mu$l of cell suspension was added into the upper chamber, whereas the lower chamber contained a volume of medium containing 10% serum ranging from 600–800 $\mu$l depending on chamber size. After incubation in an incubator for 24 h, the Transwell chambers were stained with crystal violet, and the stained cells were counted under a microscope.

## Statistical analysis

All data were replicated independently at least three times. For all bar graphs, mean ± SEM is shown. All statistical analysis was performed using GraphPad Prism 8.0. $P$-values < 0.05 were considered significant*$P$ < 0.05, **$P$ < 0.01, ***$P$ < 0.001, ****$P$ < 0.0001.

---

**Figure 4.   IDR-dependent phase separation of LEF1 is indispensable for its transcriptional activation.**
**(A)** Schematic of full-length LEF1 (FL), truncated LEF1 with IDR-depleted (Δ1, Δ12), and IDR substitution (Rec1: Δ1+fus80, Rec2: Δ12+fus162). **(B)** Activation of $\beta$-catenin with different LEF1 mutants that were detected by luciferase reporter gene assay. Error bars show the mean ± SEM from three individual experiments (***$P$ < 0.001, **$P$ < 0.01, *$P$ < 0.05, one-way ANOVA). **(C)** Co-expression of mEGFP–$\beta$-catenin and mCherry–LEF1 WT or mutant in HEK293T cells. Scale bar, 5 $\mu$m. **(D)** A quantification of the percentage of cells that showed nuclear LEF1 puncta. Data are mean ± SEM from three individual experiments. (****$P$ < 0.0001, one-way ANOVA). **(E)** Comparison of droplet formation ability about mCherry–LEF1(FL) and mutants (Δ1, Δ12, Rec 1, Rec 2) with the indicated module concentration. Scale bar, 10 $\mu$m. **(F)** Droplet formation ability for mEGFP–$\beta$-catenin with mCherry–LEF1 (FL) or with mutants mixed at a concentration of 10 $\mu$m. Scale bar, 10 $\mu$m.

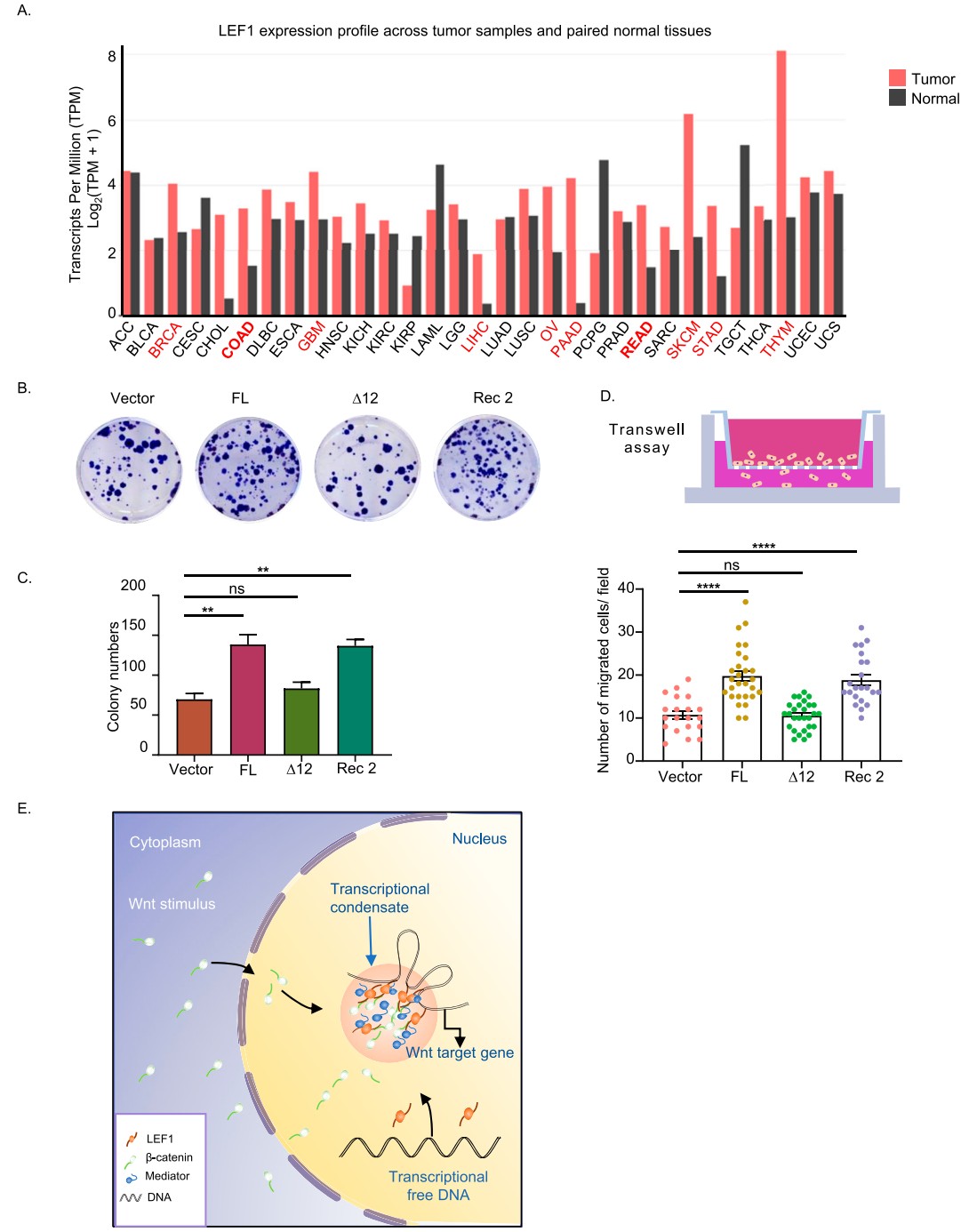

**Figure 5. LEF1 phase separation promotes tumor growth and metastasis.**
**(A)** Differences in LEF1 expression in tumor and normal tissues. The height of the bar represents the median expression of indicated tumor and normal tissues. The red font represents the top 10 greatest differences in the expression of LEF1 in tumor and normal tissues. **(B)** Representative image for colony formation of HCT116 cells after stably transfected LEF1 and its mutants, respectively. **(C)** Colony counts of each kind of HCT116 cells stably express LEF1 or its mutants. Error bars show the mean ± SEM from three individual experiments. **(D)** The statistical analysis of transwell assays of the migration in HCT116 cells. (****$P < 0.0001$, one-way ANOVA). **(E)** Model of LEF1 enhances β-catenin transactivation through IDR-dependent liquid–liquid phase separation.

# Supplementary Information

# Acknowledgements

This work was supported by grants from the National Science Fund for Excellent Young Scholars (81922051).

## Author Contributions

B Zhao: conceptualization, formal analysis, validation, and writing—original draft, review, and editing.
Z Li: resources, data curation, and writing—original draft.
S Yu: data curation and formal analysis.
T Li: data curation and methodology.
W Wang: data curation and formal analysis.
R Liu: resources, formal analysis, and methodology.
B Zhang: data curation and formal analysis.
X Fang: data curation and formal analysis.
Y Shen: data curation and formal analysis.
Q Han: data curation and methodology.
X Xu: data curation and formal analysis.
K Wang: data curation and formal analysis.
W Gong: data curation and formal analysis.
T Li: resources, data curation, and writing—original draft.
A Li: data curation and formal analysis.
T Zhou: data curation, investigation, and writing—original draft.
W Li: formal analysis, validation, project administration, and writing—original draft, review, and editing.
T Li: data curation, supervision, funding acquisition, investigation, and writing—review and editing.

## Conflict of Interest Statement

The authors declare that they have no conflict of interest.

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
