## [Reviewer comments · Life Science Alliance]

Life Science Alliance

LEF1 enhances β -catenin transactivation through IDR-dependent liquid-liquid phase separation

Bing Zhao, Zhuoxin Li, Shaoqing Yu, Tingting Li, Wen Wang, Ran Liu, Biyu Zhang, Xiya Fang, Yezhuang Shen, Qiuying Han, Xin Xu, Kai Wang, Weili Gong, Tao Li, Ai-Ling Li, Tao Zhou, Weihua Li, and Teng Li

DOI: <https://doi.org/10.26508/lsa.202302118>

Corresponding author(s): Teng Li, National Center of Biomedical Science and Weihua Li, State Key Laboratory of Proteomics, Institute of Basic Medical Sciences, China National Center of Biomedical Analysis

Review Timeline:

Submission Date:	2023-04-27
Editorial Decision:	2023-05-17
Revision Received:	2023-08-10
Editorial Decision:	2023-08-18
Revision Received:	2023-08-23
Accepted:	2023-08-23

Transaction Report:

May 17, 2023

Re: Life Science Alliance manuscript #LSA-2023-02118-T

Dr. Teng Li
National Center of Biomedical Science
cell biology
3520 San Martin Drive
Beijing 100850
China

Dear Dr. Li,

Thank you for submitting your manuscript entitled "LEF1 enhances β -catenin transactivation through IDRs-dependent liquid-liquid phase separation" to Life Science Alliance. The manuscript was assessed by expert reviewers, whose comments are appended to this letter. We invite you to submit a revised manuscript addressing the Reviewer comments.

Thank you for this interesting contribution to Life Science Alliance. We are looking forward to receiving your revised manuscript.

Sincerely,

B. MANUSCRIPT ORGANIZATION AND FORMATTING:

Reviewer #1 (Comments to the Authors (Required)):

The manuscript by Zhao and Li et. al., demonstrates that phase separation of LEF with β -catenin is necessary to drive downstream transcriptional response. Using purified proteins and in cells, authors demonstrate that LEF and β -catenin forms condensates in presence of crowding agents which demonstrate liquid like properties and punctate structures in cells, which are presumably liquid like. They further demonstrate that the IDRs in LEF drive is phase separation; IDR deletion abolishes formation of condensates in solution and in cells and it can be rescued by substitution of IDR from FUS. Phase separation is critical for driving downstream transcriptional response to Wnt signaling. Authors see that there is a high level of LEF expression in many tumors and β -catenin. They further express LEF in HCT116 cells and show that it leads to and increased proliferation and migration. While the last set of experiments are interesting there are several issues with the phase separation assays which the authors need to address before the authors can conclude and present the model as described.

1) What changes upon activation of Wnt signaling which facilitates LEF phase separation in the nucleus? Based on the results upon overexpression of β -catenin and LEF in HEK cells and in vitro experiments, it appears that LEF interaction with β -catenin is what drives co-phase separation of this proteins. However, in results presented in Fig 1 D, I do not observe a significant difference in nuclear β -catenin levels upon CHIR99021 treatment which makes me question the validity of this experiment. Secondly, I see more puncta in 1, 6 Hexanediol treatment as compared to control which seems to be contradictory to the results presented in Fig 1 A-C. For results shown in Fig 2E, what is the distribution of endogenous LEF upon β -catenin upon overexpression. This should mimic the results of CHIR99021 treatment and should be done in HEK cells or SW480 cells.

2) The materials and methods section are poorly described in relation to the phase separation assays. Only after reading the legend and not even the text, it was clear that condensate formation is seen only in presence of 10% PEG-6000. Authors should explicitly state that they need crowding reagents to induce LLPS of LEF and β -catenin. Authors should see if lowering the salt concentration to 50 or 75 mM can lead to LLPS of LEF and β -catenin without the requirement of a crowder. If use of PEG is necessary for induction of phase separation, authors should do a phase diagram by titrating PEG and protein concentration. Also, it has been previously shown that β -catenin can undergo LLPS in solution, but authors do not mention this anywhere in the text.

3) I question whether the puncta seen in cells upon overexpression of LEF or β -catenin are liquid like based on the FRAP data. First, there is no quantification provided. Second, by watching the movie there is hardly any recovery till 06:12 and a sudden increase in intensity at 06:24. This does not look like a typical FRAP recovery curve. Also, other puncta which were not bleached also show a sudden jump in intensity at these time points. Finally, the recovery only seems to be in the green channel and not the LEF channel. The FRAP experiments need to be repeated and the data must be quantified. It is possible that by 24 hours post expression the condensates could have been hardened which is fine as structures formed by LLPS can harden with time. Authors can validate the growth of condensates over time by imaging cells at earlier time points. The expectation would be that the structures grow in size over time (PMID: 32668013). Another possibility is to test for disappearance of these puncta by 1, 6 Hexanediol and reappearance post washout with appropriate controls (PMID: 33431828).

4) Results with IDR deletion and replacement with FUS IDR are impressive which stress the requirement of phase separation for activation of Wnt signaling. However, authors should show in vitro and in vivo behavior with of LEF with the deletion of the β -catenin binding domain. This will definitively confirm that interaction of LEF with β -catenin drives downstream transcriptional activation.

5) Authors have shown, that in HCT116 cells there is an increased migration and proliferation upon LEF overexpression. Authors should demonstrate the distribution of LEF and β -catenin in this background by immunostaining.

6) The materials and methods section needs to be extensively rewritten for phase separation and protein purification experiments. In case of protein purification, only references are cited. Some description about affinity tags, conditions used, whether the tags were removed should be described. The results section can have an introductory line before describing the experiments so that there is a logical flow. For example, SW480 cells come out of blue. A line about what these cells are would help the narrative. Authors should carefully proofread the manuscript as there are places where the sentence construction does not convey the correct meaning as well as correct for grammatical errors.

Referee Cross-Comments: I agree with the other reviewer that the experiments with IDR swapping is the major strength of this paper. But to conclusively demonstrate causality, the authors must show as the other reviewer suggests that binding of LEF and β -catenin is same in LLPS deficient mutants of LEF and complement it with β -catenin binding mutants of LEF, which would be competent of LLPS but not downstream transcriptional activation due to failure in co-phase separating with β -catenin.

Reviewer #2 (Comments to the Authors (Required)):

In the manuscript titled "LEF1 enhances β -catenin transactivation through IDRs-dependent liquid-liquid phase separation" the authors use in vitro droplet assays and in vivo over-expression models to show that LEF1 can form condensates and that these condensates incorporate β -catenin condensates. This condensation is dependent on N-terminal intrinsically disordered regions and can be rescued by swapping these with an IDR from FUS.

The authors present a clear and concise study where they combine a series of well executed experiments to conclude that LEF1 can form condensates together with β -catenin. The IDR-swap with FUS is particularly convincing and supports their conclusion well. However, the authors also make claims about the functionality of the co-condensation with β -catenin, while this point is only supported by a single luciferase experiment. If the authors limit their conclusions and remove the claim at causality this needs not be an issue. If the authors do want to conclude LEF1 condensate formation enhances β -catenin transactivation they need to add significant experimental work, starting with performing experiments using endogenous levels of LEF1 and β -catenin, investigating actual gene expression through RNA imaging reporters or mRNA levels (RNA-seq/qPCR). They then need to separate LEF1 mediated activation of β -catenin transactivation through LEF1 recruitment of β -catenin from condensate induced β -catenin transactivation.

Besides that major point I have a few minor questions and suggestions to improve the manuscript.

1. Grammar and English throughout the manuscript needs to be improved in some parts especially IDR (singular) versus IDRs (plural).
2. To condensate is not a verb
3. Fig1 SW480 cells have a homozygous APC Q1338* deletion which constitutively activates the Wnt pathway in these cells can the authors explain why they still use CHIR in their experiments?
4. Fig2A 20uM LEF1 shows negative space in droplet. This is not expected of in vitro homotypic droplet formation, can the authors explain why this is happening? Is this a hint at a biological function of LEF1?
5. Fig2C, 2D LEF1 droplets are found to be both salt sensitive and HD sensitive, can the authors discuss what this means for the type of interaction driving LEF1 droplet formation? Does this give a clue as to what amino acids in the LEF1 IDRs are involved?
6. Fig2F. At 5uM LEF1 and β -catenin there are non-homogeneously distributed droplets observed. Do the authors think this is relevant in vivo as well? Sub-compartmentalization has been observed for several condensates including the nucleolus where this plays an important functional role (Feric et al. 2016).
7. Figure 3G d12 seems to have an interesting core-shell structure in β -catenin. What does this mean?
8. Figure 4 displays rather striking results. Can the authors explain these results by analyzing the type of amino acids that is present in both the removed IDRs of LEF1 and the introduced IDRs of FUS that can rescue the deletions? Can the authors explain how they selected these particular domains from FUS? Is there information in these sequences that informs about the "molecular grammar" that is used for LEF1 condensation?
9. Figure 5B correlation is fully expected since LEF1 and β -catenin are WNT targets itself, and >90% has a constitutively active WNT pathway. Can the authors explain the logic of this analysis?
10. Figure 5C,D,E. These results can also be explained by β -catenin activation through conventional recruitment by LEF1 and higher activation of proliferative Wnt-target genes. To separate condensate formation from recruitment the authors need to show that the deletion mutants do not have a reduced capacity to recruit β -catenin and only affect condensate formation.

Dear Dr. Eric Sawey,

Thank you for the prompt review of our manuscript “LEF1 enhances β -catenin transactivation through IDRs-dependent liquid-liquid phase separation” by Zhao et. al. We are delighted to see that you and the reviewers find the work of interest and importance. We have addressed all of the concerns raised by the reviewers with additional data, and we provide a detailed response to their points below:

Reviewer #1

We appreciate the reviewer’s encouraging comments and constructive suggestions. We have revised manuscript by adding more data and providing in-depth discussions in accordance with these comments and suggestions. The following is our point-by-point response:

1. What changes upon activation of Wnt signaling which facilitates LEF phase separation in the nucleus? Based on the results upon overexpression of β -catenin and LEF in HEK cells and in vitro experiments, it appears that LEF interaction with β -catenin is what drives co-phase separation of this proteins. However, in results presented in Fig 1 D, I do not observe a significant difference in nuclear β -catenin levels upon CHIR99021 treatment which makes me question the validity of this experiment.

Response: We sincerely apologize for any confusion caused by the lack of clarity in presenting our data. The reviewer mentioned that the difference in nuclear β -catenin levels upon CHIR99021 treatment was not very significant. Actually, we did observe β -catenin translocation

into the nucleus following CHIR99021 treatment. In our study, in order to observe obvious puncta in the nucleus of the cell, CSK detergent extraction (+CSK), which was reported by many others for the same purpose, was used to permeabilize the cells prior to fixation and eliminate both cytoplasmic and nuclear soluble factors (PMID:21496192). Due to the CSK detergent extraction, the β -catenin translocation to the nucleus was difficult to tell at a glance. Therefore, we circled out the nucleus area by dashed lines in revised Fig 1D and showed a larger field of view (left panel, below). Meanwhile, we did a parallel experiment to show β -catenin localization following CHIR99021 treatment without CSK treatment. As shown below (right panel, below), nuclear accumulation of β -catenin was much more obvious compared to the control group. Taken together, β -catenin translocates into the nucleus and binds to LEF1 when the Wnt signaling is activated. And we show here LEF1 and β -catenin form a relatively isolated closed region through phase separation. This process ultimately results in downstream target gene transcription activation.

Secondly, I see more puncta in 1, 6-Hexanediol treatment as compared to the control which seems to be contradictory to the results presented in Fig 1 A-C.

Response: For the second question, we first apologize for our mistake in labeling the original figure 1D. The last panel should be CHIR+1,6-HD, which we have corrected in our revised paper. Secondly, we did observe punctum-like structures both in control cells and in 1,6-HD treated Wnt activated cells, however, those structures are highly variable and resistant to 1,6-HD treatment, and we didn't know the exact science and technical reality of those punctum-like structure structures yet. As shown in both Fig 1A and 1B, the larger puncta with an average area

from 0.05 to 0.15 μm^2 disappeared when treated with 1,6-HD, suggesting that these puncta formed through phase separation.

For results shown in Fig 2E, what is the distribution of endogenous LEF upon β -catenin overexpression? This should mimic the results of the CHIR99021 treatment and should be done in HEK cells or SW480 cells.

Response: Thanks to the reviewer's suggestion, we over-expressed β -catenin in HEK293T cells and showed the distribution of endogenous LEF1 in revised Supplementary data 1B. We found that in β -catenin-overexpressed cells, endogenous LEF1 exhibited a droplet pattern. These droplets were co-condensates with β -catenin. This information has been incorporated into our revised manuscript.

2. The materials and methods section is poorly described in relation to the phase separation assays. Only after reading the legend and not even the text, it was clear that condensate formation is seen only in the presence of 10% PEG-6000.

Response: We appreciate the feedback provided in the review and have incorporated additional information on 10% PEG-6000 into our revised manuscript. Furthermore, we have provided detailed descriptions of our methods in other sections.

Authors should explicitly state that they need crowding reagents to induce LLPS of LEF and β -catenin. Authors should see if lowering the salt concentration to 50 or 75 mM can lead to LLPS of LEF and β -catenin without the requirement of a crowder. If the use of PEG is necessary for induction of phase separation, authors should do a phase diagram by titrating PEG and protein

concentration. Also, it has been previously shown that β -catenin can undergo LLPS in solution, but authors do not mention this anywhere in the text.

Response: Thanks to the reviewer's suggestion, we have incorporated additional experiments and improved the manuscript accordingly. In the absence of PEG, LEF1 fails to form any condensates in lower salt concentrations of 0-150 mM (see below). We conducted titration experiments with varying concentrations of both LEF1 and PEG and included the results in the supplementary data 2C. As shown in the results below, a higher concentration of PEG leads to a lower concentration required for LEF1 to undergo LLPS. In the revised manuscript, we cited a paper that demonstrates β -catenin's ability to undergo LLPS (PMID: 31563432).

3. I question whether the puncta seen in cells upon overexpression of LEF or β -catenin are liquid-like based on the FRAP data. First, there is no quantification provided. Second, by watching the movie there is hardly any recovery till 06:12 and a sudden increase in intensity at 06:24. This does not look like a typical FRAP recovery curve. Also, other puncta which were

not bleached also show a sudden jump in intensity at these time points. Finally, the recovery only seems to be in the green channel and not the LEF channel. The FRAP experiments need to be repeated and the data must be quantified. It is possible that by 24 hours post expression the condensates could have been hardened which is fine as structures formed by LLPS can harden with time. Authors can validate the growth of condensates over time by imaging cells at earlier time points. The expectation would be that the structures grow in size over time (PMID: 32668013). Another possibility is to test for disappearance of these puncta by 1, 6 Hexanediol and reappearance post washout with appropriate controls (PMID: 33431828).

Response: Following the reviewer's suggestions, we conducted a series of experiments in addition to repeating the FRAP experiments. We conducted a time-lapse analysis for the localization of LEF1 and β -catenin after overexpression of their plasmids in cells, which revealed temporal growth of condensates. The results show that β -catenin and LEF1 formed nuclear condensates, which increased in size from 15 hrs to 17 hrs after transfection (Supplementary Data 1A). At the 18-hour time point, we replaced the cell culture medium with DMEM medium containing 5% 1,6-HD and further washed out. As expected, these condensates disappeared after 1,6-HD treatment for 3mins, and reappearance upon washout with DMEM medium. The results above were shown in revised Figure 1F.

Next, we repeated the FRAP experiment and quantified the fluorescence intensity. The results were shown in the revised Supplementary Data 1C,1D and below.

4. Results with IDR deletion and replacement with FUS IDR are impressive which stress the requirement of phase separation for activation of Wnt signaling. However, authors should show *in vitro* and *in vivo* behavior of LEF with the deletion of the β-catenin binding domain. This will definitively confirm that interaction of LEF with β-catenin drives downstream transcriptional activation.

Referee Cross-Comments: I agree with the other reviewer that the experiments with IDR

swapping is the major strength of this paper. But to conclusively demonstrate causality, the authors must show as the other reviewer suggests that binding of LEF and β -catenin is same in LLPS deficient mutants of LEF and complement it with β -catenin binding mutants of LEF, which would be competent of LLPS but not downstream transcriptional activation due to failure in co-phase separating with β -catenin.

Response: We greatly appreciate the reviewer's agreement with IDR swapping is major strength of this work. Following the review's suggestion, we conducted an *in vivo* experiment to investigate the localization of LEF1 mutant with β -catenin binding region deletion. We observed that a deletion mutant lacking the β -catenin binding domain of LEF1 exhibited significantly reduced ability to form condensates. Additionally, overexpression of this LEF1 mutant and mEGFP- β -catenin led to decrease of β -catenin foci in nuclear. It is already known that LEF1 isoform lacking the N-terminal high affinity β -catenin binding domain acts as a dominant negative variant to inhibit full-length LEF1 function in colorectal cancer (PMID: 17251379, PMID: 17143293). Therefore, these findings suggest that the interaction between LEF1 and β -catenin plays an important role in their co-condensation formation and transactivation. The results were shown in Supplementary Data 3A,3B. For studying the function LEF1 with β -catenin binding domain deletion mutant *in vitro*, we tried to purify LEF1 mutant using Sf9 insect cells. Unfortunately, our attempts to purify eukaryotic proteins *in vitro* did not yield sufficient purity for further experiments.

5. Authors have shown, that in HCT116 cells there is an increased migration and proliferation upon LEF overexpression. Authors should demonstrate the distribution of LEF and β -catenin in this background by immunostaining.

Response: We thank the reviewer for their valuable and constructive suggestion. We

overexpressed LEF1 wide type and mutants in HCT116 cells and observed the localization of LEF1 and β -catenin by immunostaining. However, we failed to find any LEF1 condensates and specific β -catenin staining in nuclear. We presume that the endogenous β -catenin level is quite low compared to the overexpressed LEF1 in HCT116 cell. We believe that the foci condensation of LEF1 only can be seen under β -catenin overexpression. Therefore, the LEF1 and β -catenin condensates formation were difficult to observe though sufficient to promote cell growth and migration.

6. The materials and methods section needs to be extensively rewritten for phase separation and protein purification experiments. In case of protein purification, only references are cited. Some description about affinity tags, conditions used, whether the tags were removed should be described. The results section can have an introductory line before describing the experiments so that there is a logical flow. For example, SW480 cells come out of blue. A line about what these cells are would help the narrative. Authors should carefully proofread the manuscript as there are places where the sentence construction does not convey the correct meaning as well as correct for grammatical errors.

Response: We thank the reviewer for this criticism and have revised our manuscript. And we have provided a more detailed description of the results and methods, and we corrected the grammatical errors in revised paper.

Reviewer #2

The reviewer commended our work for presenting a clear and concise study on the ability of LEF1 to form condensates with β -catenin. Additionally, the reviewer found our IDR swap with FUS to be particularly convincing and supportive of our conclusion. However, the reviewer noted that the functionality of co-condensation with β -catenin was only supported by a single luciferase experiment.

Response: We appreciate the reviewer's constructive criticism and have incorporated additional experiments to support our conclusions. First, we examined the mRNA expression of Wnt target genes in HCT116 cells transfected with LEF1 and IDR deletion mutants. Our findings demonstrate that wild-type LEF1 enhances c-myc and c-Jun mRNA expression, while the phase separation deficient mutant does not (see below). We have added this data in the revised manuscript (Supplementary Data 4D).

Additionally, we conducted an *in vivo* experiment to investigate the localization of LEF1 mutant with β -catenin binding region deletion. We observed that a deletion mutant lacking the β -catenin binding domain exhibited significantly reduced ability to form condensates. Interestingly, overexpression of this LEF1 mutant and mEGFP- β -catenin also led to decrease of β -catenin foci in nuclear. It is already known that LEF1 isoform lacks the N-terminal high affinity β -catenin binding domain and acts as a dominant negative variant to inhibit full-length LEF1 function in colorectal cancer (PMID: 17251379, PMID: 17143293). Therefore, all the above results support that the co-condensation of LEF1 with β -catenin promotes the β -catenin mediated transactivation in colorectal cancer cells. The results were shown in below and Supplementary Data 3A,3B.

1. Grammar and English throughout the manuscript need to be improved in some parts especially IDR (singular) versus IDRs (plural).

Response: We thank the reviewer for the critical reading. We asked a native English speaker to limit grammar errors.

2. To condensate is not a verb.

Response: Sorry for that mistake. It has been corrected along with other grammatic errors in the revision.

3. Fig1 SW480 cells have a homozygous APC Q1338* deletion which constitutively activates the Wnt pathway in these cells can the authors explain why they still use CHIR in their experiments?

Response: We thank the reviewer for bringing attention to the significant characteristic of the SW480 cell line. At the onset of our investigation, we opted to examine cells exhibiting a comparatively elevated expression of LEF1, such as SW480, in order to observe the dynamic condensate formation of LEF1 *in vivo*. We treated cells with Wnt activator CHIR99021 and found a notable increase in the translocation of β-catenin into the nucleus (see below). Meanwhile, the β-catenin protein level didn't change much upon the treatment. We are not 100% sure about the exact mechanism of this phenomenon. Previous research has indicated that the phosphorylation of β-catenin by GSK-3 plays a crucial role in determining its degradation via the ubiquitin-proteasome system. Additionally, the phosphorylation event also governs the translocation of β-catenin from the cytoplasm to the nucleus (PMID:21496192), as well as its interactions with various other molecules. Hence, the potential enhancement of nuclear

translocation of β -catenin through the inhibition of glycogen synthase kinase-3 (GSK-3) by CHIR99021 can be attributed to its ability to impede the phosphorylation of β -catenin and its interaction with other proteins. It is also possible that the inhibition of GSK-3 β leads to direct or indirect posttranslational modifications of specific LEF/TCFs, thereby causing differential regulation and expression of these factors in various tissues and cell lines.

4. Fig2A 20uM LEF1 shows negative space in the droplet. This is not expected of in vitro homotypic droplet formation, can the authors explain why this is happening? Is this a hint at a biological function of LEF1?

Response: We thank the reviewer for the thorough observation. We have also observed the phenomenon of vacuoles appearing in droplets with an increased concentration of LEF1 in our experiment. In some literature, similar phenomena were observed (PMID: 27212236). The authors propose that these phenomena are caused by differences in the biophysical properties of the phases and suggest that phase separation can give rise to multilayered liquids, which may facilitate sequential RNA processing reactions in a variety of RNP bodies. This phenomenon of LEF1 may also be related to its transcriptional function, which needs further investigation.

5. Fig2C, 2D LEF1 droplets are found to be both salt sensitive and HD sensitive, can the authors discuss what this means for the type of interaction driving LEF1 droplet formation? Does this give a clue as to what amino acids in the LEF1 IDRs are involved?

Response: The phase separation of LEF1 protein is influenced by a high concentration of salt and 1,6-HD, which is possibly due to electrostatic interaction and hydrophobic interaction of molecular interaction both contribute to the formation of the droplets. Therefore, our analysis reveals that a variety of molecular interactions contribute to LEF1 condensate formation. We

observed a high content of aromatic amino acids in the IDR region of LEF1, suggesting that p- π conjugation may be involved in phase separation. To investigate this, we mutated Y and F residues to S in the IDR region of LEF1. However, we found the Y/F to S mutant can still undergo phase separation *in vivo*. Thus far, we have not identified what amino acids contribute to LEF1 condensate formation. Thank you again for your valuable suggestions during the review process.

6. Fig2F. At 5 μ M LEF1 and β -catenin there are non-homogeneously distributed droplets observed. Do the authors think this is relevant *in vivo* as well? Sub-compartmentalization has been observed for several condensates including the nucleolus where this plays an important functional role *in vivo*.

Response: Thank you for your thorough observation. At a concentration of 5 μ M, LEF1 and β -catenin exhibit heterogeneous droplet distribution, which may have implications for their functional roles. The distinct surface tension (PMID: 27212236) of these two proteins potentially supports their function at specific concentrations, thereby influencing sequential DNA processing steps. However, we have not yet observed consistent phenomena that replicate those observed *in vivo*. Hence, we will persist in monitoring this phenomenon and conducting comprehensive research.

7. Figure 3G d12 seems to have an interesting core-shell structure in β -catenin. What does this mean?

Response: Thank you very much for the reviewer's thorough observation. We have carefully examined the original data and regret to inform that the core-shell structure in β -catenin is atypical, possibly due to uneven mixing of the two proteins during experimental operation. As such, we have substituted the image with one without the structure, which should be more representative in our revised manuscript.

8. Figure 4 displays rather striking results. Can the authors explain these results by analyzing the type of amino acids that is present in both the removed IRDs of LEF1 and the introduced IDRs of FUS that can rescue the deletions? Can the authors explain how they selected these particular domains from FUS? Is there information in these sequences that informs about the "molecular grammar" that is used for LEF1 condensation?

Response: Firstly, we observed a very limited homology, less than 20%, between the intrinsically disordered region (IDR) of LEF1 and FUS, indicating minimal functional similarity between them (see below). Furthermore, these two IDRs exhibited distinct amino acid enrichment profiles, LEF1 IDR is enriched with Y and F, and FUS with QGSY- and G-rich domains. Additionally, previous studies have utilized FUS IDRs as substitutes for other protein IDRs to investigate liquid-liquid phase separation (LLPS) (PMID:29650702, PMID: 34163069, PMID: 34038708). Therefore, we selected the IDR region of FUS as a replacement for the LCD domain in LEF1 to study its phase separation functionality. So far, we have not found characteristics of these sequences as the “molecular grammar” for LEF1 condensation formation.

9. Figure 5B correlation is fully expected since LEF1 and β -catenin are WNT targets themselves, and >90% has a constitutively active WNT pathway. Can the authors explain the logic of this analysis?

Response: We appreciate the reviewer's suggestion, we agree that the correlation was fully expected since LEF1 and β -catenin is WNT targets themselves. Following the advice of the reviewer, we have transferred the data to the revised supplementary materials(supplementary data 4C).

10. Figure 5C,D,E. These results can also be explained by β -catenin activation through conventional recruitment by LEF1 and higher activation of proliferative Wnt-target genes. To separate condensate formation from recruitment the authors need to show that the deletion mutants do not have a reduced capacity to recruit -catenin and only affect condensate formation.

Response: We apologize for the lack of clarity in presenting our data. The previous supplementary data included an experiment on the binding ability of LEF1 deletion mutant to β -catenin, which we have now incorporated into the main figure (Figure 3 F) based on the reviewer's suggestion.

Sincerely yours,

Wei-Hua Li, Ph.D.

Teng Li, Ph.D.

Associate Professor

National Center of Biomedical Analysis

EMAIL: whli@ncba.ac.cn; tnli@ncba.ac.cn

August 18, 2023

RE: Life Science Alliance Manuscript #LSA-2023-02118-TR

Dr. Teng Li
National Center of Biomedical Science
cell ciology
No.27, Taiping Road
Beijing 100850
China

Dear Dr. Li,

Thank you for submitting your revised manuscript entitled "LEF1 enhances β -catenin transactivation through IDR-dependent liquid-liquid phase separation". We would be happy to publish your paper in Life Science Alliance pending final revisions necessary to meet our formatting guidelines.

- please address Reviewer 1's remaining minor comments
- please upload all figure files as individual ones, including the supplementary figure files; all figure legends should only appear in the main manuscript file
- please add ORCID ID for the 2ndary corresponding author--they should have received instructions on how to do so
- please add a Summary Blurb/Alternate Abstract to our system
- please add the Twitter handle of your host institute/organization as well as your own or/and one of the authors in our system
- please consult our manuscript preparation guidelines <https://www.life-science-alliance.org/manuscript-prep> and make sure your manuscript sections are in the correct order
- please update the supplementary figures in the system to Supplementary Figures S1, S2, etc.. and ensure that the figures are named correctly.

Figure Check:

- please add sizes next to blots in Figure 3F

A. FINAL FILES:

B. MANUSCRIPT ORGANIZATION AND FORMATTING:

Sincerely,

Reviewer #1 (Comments to the Authors (Required)):

The manuscript by Zhao et al., which addresses an important question regarding LEF1 LLPS in regulating transcriptional activation downstream of Wnt signaling and hence a potential therapeutic angle has been significantly improved in the revised form. Authors have made a commendable job of addressing and clarifying all my experimental concerns and those raised by the other reviewer while also significantly improving the writing style while presenting their results and details of the experiments in the materials and methods section. I therefore happily recommend the publication of this important piece of work.

Minor:

I encourage the authors to go through the text once again as I thought minor changes might be required in a few places. I am highlighting two of them.

Line nos 72-76 can be reconstructed as follows if authors find this appropriate: In our previous studies we noticed that LEF1 exhibited punctate distribution in SW40, Jurkat and HEK293T cell when expressed at high levels. This is reminiscent of proteins forming condensates in cells via liquid-liquid phase separation (Ahn et al, 2021).

Line no 158 nuclear should be replaced by nucleus. Include the Figure numbers describing the results after nucleus in line 158 and delete them from line no 160.

Reviewer #2 (Comments to the Authors (Required)):

Based on my earlier comments and also in response to the other reviewer's comments the authors have significantly improved the manuscript. They have improved and clarified the work supporting the LLPS-driven condensate formation of LEF1 and β -catenin. They have also added experimental data to support a causal link between condensate formation and β -catenin/LEF1 driven transcription. While the full extent of causal relation between transcription factor condensate formation and active transcription needs to be further elucidated, this study convincingly observes β -catenin/LEF1 co-condensation coupled to active transcription. I therefore support publication of this manuscript in Life Science Alliance.

Dear Editor,

Thank you very much for your kind efforts and consideration in facilitating the publication of our paper. We have addressed the review1's comments point by point below.

Reviewer #1 (Comments to the Authors (Required)):

The manuscript by Zhao et al., which addresses an important question regarding LEF1 LLPS in regulating transcriptional activation downstream of Wnt signaling and hence a potential therapeutic angle has been significantly improved in the revised form. Authors have made a commendable job of addressing and clarifying all my experimental concerns and those raised by the other reviewer while also significantly improving the writing style while presenting their results and details of the experiments in the materials and methods section. I therefore happily recommend the publication of this important piece of work.

Minor:

I encourage the authors to go through the text once again as I thought minor changes might be required in a few places. I am highlighting two of them.

Line nos 72-76 can be reconstructed as follows if authors find this appropriate: In our previous studies we noticed that LEF1 exhibited punctate distribution in SW40, Jurkat and HEK293T cell when expressed at high levels. This is reminiscent of proteins forming condensates in cells via liquid-liquid phase separation (Ahn et al, 2021).

Line no 158 nuclear should be replaced by nucleus. Include the Figure numbers describing the results after nucleus in line 158 and delete them from line no 160.

Response: We are very appreciated for review's suggestions. We have made modifications to the manuscript based on Reviewer 1's suggestion.

First, we believe that the modification from reviewer is more reasonable, and we have replaced the original description with a new one. (line nos 72-76 of previous revision manuscript)

Secondly, we have replaced the "nuclear" to "nucleus" (line no 158 of previous revision manuscript). And we also moved the figure numbers after the "nucleus" (from line 160 to the line 158 of previous revision manuscript).

Sincerely yours,

Wei-Hua Li, Ph.D.

Teng Li, Ph.D.

Associate Professor

National Center of Biomedical Analysis

EMAIL: whli@ncba.ac.cn;

tnli@ncba.ac.cn

August 23, 2023

RE: Life Science Alliance Manuscript #LSA-2023-02118-TRR

Dr. Teng Li
National Center of Biomedical Science
cell ciology
No.27, Taiping Road
Beijing 100850
China

Dear Dr. Li,

Thank you for submitting your Research Article entitled "LEF1 enhances β -catenin transactivation through IDR-dependent liquid-liquid phase separation". It is a pleasure to let you know that your manuscript is now accepted for publication in Life Science Alliance. Congratulations on this interesting work.

DISTRIBUTION OF MATERIALS:

Again, congratulations on a very nice paper. I hope you found the review process to be constructive and are pleased with how the manuscript was handled editorially. We look forward to future exciting submissions from your lab.

Sincerely,
